# Incidental Detection of Ovarian Cancer as a Solitary Peridiaphragmatic Distant Lymph Node Metastasis without Pelvic Lesions on ^18^F-FDG PET/CT

**DOI:** 10.3390/diagnostics11030422

**Published:** 2021-03-02

**Authors:** Hye Joo Son, Yong-Moon Lee, Jai-Hyuen Lee

**Affiliations:** 1Department of Nuclear Medicine, Dankook University College of Medicine, 201 Manghyang-ro, Dongnam-gu, Cheonan, Chung Nam 31116, Korea; hyejooson@dkuh.co.kr; 2Department of Pathology, Dankook University College of Medicine, 201 Manghyang-ro, Dongnam-gu, Cheonan, Chung Nam 31116, Korea; vilimoon@dkuh.co.kr

**Keywords:** ovarian serous papillary carcinoma, oligometastases, peridiaphragmatic lymph node, ^18^F-FDG PET/CT

## Abstract

The spreading pattern of ovarian carcinoma is unique and unlike most other cancers, because exfoliated ovarian cancer cells primarily disseminate within the abdominal cavity, which are then transported throughout the peritoneum by physiological peritoneal fluid. An initial manifestation of a solitary peridiaphragmatic distant metastatic lymph node without peritoneal involvement is very rare. This study reports a case with an incidentally found single hypermetabolic mass in the peridiaphragmatic space without a pelvic lesion in the baseline staging ^18^ F-FDG PET/CT that histologically turned out to be metastatic serous papillary carcinoma due to ovarian cancer. ^18^F-FDG PET/CT may allow the identification of the initial manifestation of unexpected distant oligometastatic statuses of an unknown primary ovarian cancer.

## Figures and Tables

**Figure 1 diagnostics-11-00422-f001:**
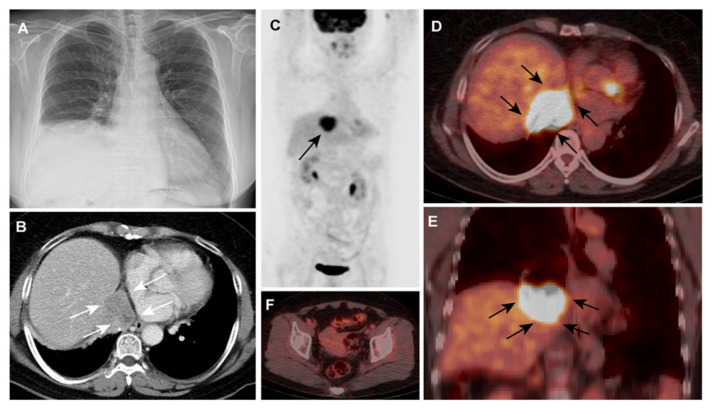
A 67-year old woman was admitted to our hospital for the elevation of right hemidiaphragm that was incidentally found by chest radiography at a regular checkup (Figure 1, Panel A). Her medical history was unremarkable (e.g., no trauma or surgery), and she had no specific symptoms. A thoraco-abdominal enhanced computed tomography (CT) scan revealed the lobulated heterogeneous enhancing enlarged lymph node that was 4.2 cm in diameter and in the right posterior cardiophrenic space with a directly abutting liver dome and left atrium; the node was invading the right phrenic nerve, causing the elevation of the right hemidiaphragm (Figure 1B, arrows). The CA-125 value was 142.9 U/mL (normal range, 0–35 U/mL [1]). On the same day, ^18^F-FDG PET/CT imaging was conducted (blood glucose level, 84 mg/dl; injected dose: 5.98 mCi). ^18^F-FDG PET/CT imaging MIP (Figure 1C, arrow), axial fused PET/CT (Figure 1D, arrow), and coronal fused PET/CT (Figure 1E, arrow) revealed an intense hypermetabolic mass (SUVmax 9.5) in the abovementioned sites. There were no significant hypermetabolic lesions in the pelvic cavity suggesting primary malignant lesions or pelvic metastases (Figure 1F).

**Figure 2 diagnostics-11-00422-f002:**
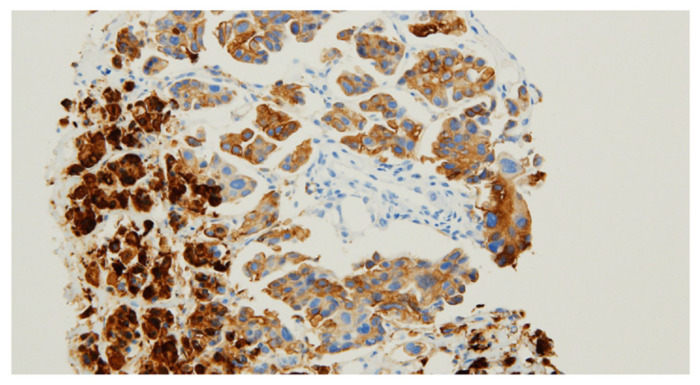
Percutaneous needle biopsy was conducted in the right diaphragm, and the specimen was submitted for histopathological examinations (Figure 2). To exclude the possibility of metastatic papillary carcinoma from the lung or thyroid origins and papillary mesothelioma, the immunohistochemical stains for TTF-1, napsin-A, thyroglobulin, calretinin, and D2-40 were performed, but none were reactive to the tumor cells. Finally, tumor cells were positive with the paired-box gene 8 (PAX8), which led to the diagnosis of ovarian serous papillary carcinoma (A) [2]. Ovarian cancers are usually confined to the abdominopelvic cavity because they spread mainly via the direct extension or peritoneal surface implantation followed by intraperitoneal lymphatic spread [3,4]. Metastases to intrathoracic lymph nodes are infrequent, reporting a low rate of 2.3% (six out of 255 cases) [5]. Metastases of ovarian carcinoma to intrathoracic lymph nodes tend to follow a predictable pattern: the afferent lymphatics drain from the diaphragm, liver, pleura, and anterior abdominal wall and empty into the internal mammary chain [6]. The natural spreading pattern of ovarian cancer includes extensive tumor dissemination on the peritoneal and pleural spaces first before intrathoracic lymph node metastases [7]. Therefore, the manifestation of isolated distant lymph node metastasis without any indication of primary malignant lesions or seeding metastases in the pelvic cavity is very rare during the initial staging setting [8]. Possible pathways for tumor spread in our case are still questionable, considering that the serosal surfaces and intraabdominal viscera were unaffected. Isolated intrathoracic lymph node metastases is known to be an important predictor of overall survival, resulting in advanced stages and poor prognosis [7,9].^18^F-FDG PET/CT may identify the initial manifestation of unexpected distant oligometastatic statuses of an unknown primary ovarian cancer.

## Data Availability

The data presented in this study are available upon request from the corresponding author.

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
