# Peer review of "Incidental Detection of Ovarian Cancer as a Solitary Peridiaphragmatic Distant Lymph Node Metastasis without Pelvic Lesions on 18F-FDG PET/CT"

_diagnostics, 2021, doi:10.3390/diagnostics11030422_

Round 1
Reviewer 1 Report
Dear Authors,
This case of incidental finding of distant lymph node metastasis in ovarian cancer is interesting and well described. Thanks for sharing your experience.
Main comments:
L31 Please add a reference to the normal range values.
L31-35 Please add the elapsed time between CT and PET/CT, the patient blood glucose level and the injected activity.
L55-56 Please, add the rationale, on the basis of the CARE statement (doi: 10.7453/gahmj.2013.008):
- Rationale for conclusions (including assessments of cause and effect)
Author Response
Reviewer 1:
This case of incidental finding of distant lymph node metastasis in ovarian cancer is interesting and well described. Thanks for sharing your experience.
Comment 1: L31 Please add a reference to the normal range values.
Authors’ response:
Thank you for the comment. We corrected the normal range of CA-125 to 0–35 U/mL and added the reference regarding the reviewer's suggestion.
In the L32 of the manuscript:
The CA-125 value was 142.9 U/mL (normal range, 0–35 U/mL [1]).
References
- Bast Jr, R.C.; Klug, T.L.; John, E.S.; Jenison, E.; Niloff, J.M.; Lazarus, H.; Berkowitz, R.S.; Leavitt, T.; Griffiths, C.T.; Parker, L. A radioimmunoassay using a monoclonal antibody to monitor the course of epithelial ovarian cancer. New England journal of medicine 1983, 309, 883-887.
Comment 2: L31-35 Please add the elapsed time between CT and PET/CT, the patient blood glucose level and the injected activity.
Authors’ response:
CT and PET/CT was conducted on the same day. Blood glucose level was 84 mg/dl and injected activity was 5.98 mCi. We added the information as the reviewer suggested.
In the L31-32 of the manuscript:
On the same day, 18F-FDG PET/CT imaging was conducted (blood glucose level, 84 mg/dl, injected dose: 5.98 mCi).
Comment 3: L55-56 Please, add the rationale, based on the CARE statement (doi: 10.7453/gahmj.2013.008): Rationale for conclusions (including assessments of cause and effect)
Authors’ response:
Thank you so much for your insightful comment. By integrating FDG PET/CT imaging finding and pathological analysis, we concluded that we presented a very rare case of solitary, peridiaphragmatic metastatic lymph node, which was histologically turned out to be metastatic serous papillary carcinoma due to ovarian cancer, without a primary pelvic lesion in the baseline staging 18 F-FDG PET/CT. On the additional histologic image of the biopsied tumor with microcalcification (in red circle), we identified a typical papillary structure resembling a cross-sectioned finger, which is the pathologic signature of papillary carcinoma. Because papillary carcinoma with microcalcification can be developed from other organs, such as thyroid or mesothelium, we excluded other possibilities of origins by investigating all possible organ-specific protein expressions. TTF-1, napsin-A, thyroglobulin, calretinin, and D2-40 were performed, but none were reactive to the tumor cells. Strong expression of PAX8, ovarian specific protein marker, supports the conclusion that solitary peridiaphragmatic lymph node in our case is originated from ovary. So, 18F-FDG PET/CT is useful for the identification of the initial manifestation of unexpected distant oligometastatic statuses of an unknown primary ovarian cancer. Furthermore, I added CARE checklist for my article.

Reviewer 2 Report
The authors present a rare case of solitary peridiaphragmatic distant metastatic lymph node without peritoneal involvement in a patient with ovarian carcinoma. FDG PET imaging contributed to the identification of the lesion, which was subsequently analyzed by means of a percutaneous needle biopsy.
The case is well presented and the images are clear and suggestive.
No major or minor revisions are required.
Author Response
We appreciate the reviewer’s positive comments and understanding of our paper.
